# Cardiovascular Complications of Obstructive Sleep Apnea in the Intensive Care Unit and Beyond

**DOI:** 10.3390/medicina58101390

**Published:** 2022-10-03

**Authors:** Abdul Wahab, Arnab Chowdhury, Nitesh Kumar Jain, Salim Surani, Hisham Mushtaq, Anwar Khedr, Mikael Mir, Abbas Bashir Jama, Ibtisam Rauf, Shikha Jain, Aishwarya Reddy Korsapati, Mantravadi Srinivasa Chandramouli, Sydney Boike, Noura Attallah, Esraa Hassan, Mool Chand, Hasnain Saifee Bawaadam, Syed Anjum Khan

**Affiliations:** 1Department of Hospital Medicine, Mayo Clinic Health System, Mankato, MN 56001, USA; 2Section of Hospital Medicine, Hospital of the University of Pennsylvania, Philadelphia, PA 19104, USA; 3Department of Critical Care Medicine, Mayo Clinic Health System, Mankato, MN 56001, USA; 4Department of Medicine and Pharmacology, Texas A&M University, College Station, TX 79016, USA; 5Department of Internal Medicine, St Vincent’s Medical Center, Bridgeport, CT 06606, USA; 6Department of Internal Medicine, BronxCare Health System, Bronx, NY 10457, USA; 7Department of Medicine, University of Minnesota, Minneapolis, MN 55455, USA; 8Department of Medicine, St. George’s University School of Medicine, St. George SW17 0RE, Grenada; 9Department of Medicine, MVJ Medical College and Research Hospital, Karnataka 562114, India; 10Department of Medicine, University of Buckingham Medical School, Buckingham MK18 1EG, UK; 11Department of Medicine, Trust Hospital, Kakinada 533005, India; 12Department of Pulmonary & Critical Care Medicine, Aurora Medical Center, Kenosha, WI 53140, USA

**Keywords:** Intensive Care Units (ICU), mortality, obesity, sleep apnea, obstructive sleep apnea (OSA), sleep apnea syndrome (SAS), cardiovascular system, atrial fibrillation, stroke, coronary artery disease, outcomes, arrythmia, pulmonary hypertension (PH), CPAP, treatment effective

## Abstract

Obstructive sleep apnea (OSA) is a common disease with a high degree of association with and possible etiological factor for several cardiovascular diseases. Patients who are admitted to the Intensive Care Unit (ICU) are incredibly sick, have multiple co-morbidities, and are at substantial risk for mortality. A study of cardiovascular manifestations and disease processes in patients with OSA admitted to the ICU is very intriguing, and its impact is likely significant. Although much is known about these cardiovascular complications associated with OSA, there is still a paucity of high-quality evidence trying to establish causality between the two. Studies exploring the potential impact of therapeutic interventions, such as positive airway pressure therapy (PAP), on cardiovascular complications in ICU patients are also needed and should be encouraged. This study reviewed the literature currently available on this topic and potential future research directions of this clinically significant relationship between OSA and cardiovascular disease processes in the ICU and beyond.

## 1. Introduction

Obstructive sleep apnea (OSA) is a disease that causes recurring episodes of decreased airflow while sleeping due to episodic collapse of the upper airway, despite adequate respiratory effort. This leads to a drop in oxyhemoglobin saturation, an increase in intrathoracic pressure, and bursts of sympathetic activity causing increased heart rate and blood pressure [1,2,3]. In a symptomatic patient, OSA is defined by at least five obstructive events (apneas, hypopneas) per hour through sleep [4]. This results in sleep disturbance leading to excessive daytime sleepiness, mood changes, decreased concentration, and fatigue [4]. Worldwide, OSA is reported to affect 5–25% of adults and has been shown to independently increase the risk for several cardiovascular diseases, such as hypertension, heart failure, ischemic heart disease, and cardiac arrhythmias [5,6,7]. OSA is more common in the obese, male gender, and elderly individuals, with additional risk factors being ethnicity, family history, and altered craniofacial anatomy [8,9,10]. Ironically, women with OSA have a higher incidence of congestive heart failure (CHF) and increased biomarkers, such as troponin (Tn), compared to men [9,11].

OSA prevalence in patients admitted to ICUs is estimated to be around 4–8% [6,9,12]. Most patients admitted to an ICU usually have pulmonary and cardiac comorbid conditions [13]. OSA in these patients can potentially complicate their critical illness. Patients with obesity and underlying sleep-related breathing disorders are at increased risk of hypercapnic respiratory failure [14]. The use of nocturnal PAP therapy in non-ICU heart failure patients has been shown to improve left ventricular systolic function by reducing apnea/hypopnea events [1]. OSA patients are more likely to require ICU admission and non-invasive or invasive ventilation after surgery. Ironically, some studies reported reduced ICU and hospital mortality in ICU patients; however, others did not find such benefits [6,12,15]. The effect of OSA on the outcomes of ICU patients remains controversial. In this study, we reviewed the current literature on the impact of OSA on certain cardiovascular diseases, mortality, and length of stay in patients admitted to the ICU.

## 2. Atrial Fibrillation

OSA is highly prevalent in patients admitted to the hospital with cardiovascular diseases. It is associated with a number of cardiac rhythm disturbances, such as bradycardia, sinus pauses, atrial fibrillation (AF), ventricular ectopy, ventricular fibrillation/tachycardia (VF/VT), and sudden cardiac death (SCD) [16].

Atrial fibrillation (AF) is the most common arrhythmia in ICU patients, with an incidence of 5–50% in patients with severe sepsis [17,18]. Based on limited retrospective data, the prevalence of OSA is about 4–8% of ICU patients and 42–56% in patients with AF [6,9,19]. In the non-ICU patient population, the correlation between OSA and AF has been studied extensively, and both conditions were found to coexist frequently, but causality was not well-established [20].

Evidence suggests that inflammation during critical illness leads to an accelerated cardiac structural and electrical remodeling, which makes the heart susceptible to new-onset AF or recurrence of AF and rapid ventricular rate [21]. Like other organ dysfunctions, new onset AF with or without rapid ventricular rate is considered a marker of disease severity during critical illness. Theoretically, OSA in critically ill patients could be an additional trigger for AF. The impact of OSA on AF and other cardiac arrhythmias is a matter of ongoing study in critically ill patients. The few studies that have reported the effects of OSA on mortality, length of stay, and postoperative outcomes in critically ill patients, unfortunately, did not report data on cardiac arrhythmia [6,12,15,22].

In the non-ICU patient population, published data suggest that continuous positive pressure airway (CPAP) in OSA patients with AF reduces AF recurrence after ablation or cardioversion [23]. However, a small randomized controlled trial consisting of twenty-five non-sleepy mild to moderate OSA patients could not replicate these positive results [24].

Likewise, a study by Abe et al. which included 316 patients with various cardiac arrhythmias demonstrated that OSA treatment with CPAP decreases the burden of cardiac arrhythmia, especially that of paroxysmal AF (PAF) in Japanese patients [25]. However, a randomized controlled trial by Traaen et al., which included 108 patients with moderate to severe OSA, did not show any significant impact of CPAP therapy on the burden of PAF [26].

## 3. Non-AF Arrhythmias

Other arrhythmias that include bradycardia, tachycardia, a combination of tachycardia and bradycardia, sinus pauses, ventricular ectopy, ventricular tachycardia, trigeminy, bigeminy, as well as variable degrees of atrio-ventricular blocks, including asystole, can occur in patients with OSA [27,28,29]. Many of these are caused due to a combination of initial hypoxia followed by hypercapnia during the apnea-hypopnea events, associated with varying intra-thoracic pressure and arousal mechanisms, resulting in sleep fragmentation [27,30,31]. This leads to carotid body stimulation causing increased vagal or parasympathetic tone followed by sympathetic stimulation that follows a period of decreased oxygen, increased carbon dioxide, and increased negative intrathoracic pressure [27,30,31]. Hypoxia may also cause slower depolarization and increased automaticity due to increased activity of the sympathetic-parasympathetic system. As a result, OSA along with accompanying hypoxia is a potent stimulus for several types of arrhythmias [28,29,30,31].

In a population-based study involving more than 10,000 patients, moderate to severe OSA and its associated nocturnal hypoxia were independent risk factors for sudden cardiac death (SCD) [27]. The mechanism that could be attributed to SCD is ventricular arrhythmia [27]. A proof-of-concept study is the association of prolonged QT intervals noted in patients with congenital long QT syndrome who had OSA compared to those who did not have OSA [32]. Long QT intervals adjusted for heart rate have been associated with an increased risk for SCD and VT/VF [32]. Treatment with PAP therapy has been shown to suppress arrhythmias such as ventricular premature beats, ventricular bradycardia, and asystole, though it is unknown if it can help suppress other types of arrhythmias or improve overall survival [33,34,35].

## 4. Heart Failure

Both OSA and central sleep apnea (CSA) are common in heart failure [36,37]. The prevalence of OSA in patients with heart failure varies from 47–81% [36,37]. OSA is an independent risk factor for heart failure and can accelerate the progression to acute decompensation [1,3]. As compared to the general population, OSA is much more prevalent in patients with heart failure with reduced ejection fraction (12–53%) and patients with heart failure with preserved ejection fraction (35–64%) [37].

Both OSA and heart failure exhibit common mechanisms that worsen cardiac function. OSA, as noted above, is characterized by increased sympathetic activity both during sleep and while being awake [36]. This results in increased heart rate and blood pressure along with arrhythmia [36]. Furthermore, the increased negative intrathoracic pressure caused by OSA leads to increased preload (by increasing venous return and causing right ventricular enlargement and decreasing left ventricular volume) and increased afterload (by adversely affecting transmural pressure) [36]. This results in decreased stroke volume. Conversely, severe heart failure can affect upper airway patency by the rostral displacement of fluid accumulated during the day in the extremities during sleep, resulting in pharyngeal edema and venous congestion, and worsening the underlying OSA [38]. Furthermore, patients with severe heart failure have significant respiratory control instability due to chronic hyperventilation and tightly regulated arterial carbon dioxide levels, and this adds to upper airway obstructive events in patients with underlying OSA [39].

Nocturnal hypoxia has been shown to cause a unique left ventricular remodeling pattern and is an independent predictor of mortality in heart failure patients [40,41,42]. Other than its direct effect on heart failure, OSA causes or worsens heart failure by its impact on often co-existing hypertension, cardiac arrhythmias, and coronary artery disease [43]. Studies have shown that CPAP use in patients with heart failure and OSA reduces sympathetic activity, blood pressure, ventricular arrhythmia, and improves gas exchange and left ventricular ejection fraction [44,45]. Chronic use of CPAP over weeks to months leads to significant improvement in left ventricular filling pressure and ejection fraction [1,46]. Such improvement in left ventricular function gets worse on withdrawal of CPAP even just for one week, suggesting a significant role of OSA on severity of heart failure [46]. In a randomized controlled trial involving 258 patients with OSA and heart failure, the use of PAP therapy resulted in improved LV function, improved walking distance, decreased norepinephrine, improved nocturnal oxygenation, improved apnea hypopnea index (AHI), but no change in survival compared to usual care [36]. Regardless of OSA, CPAP is an effective treatment for respiratory distress in patients with cardiogenic pulmonary edema, and decreases the need for mechanical ventilation [47]. Therefore, in patients with acute decompensated heart failure and underlying OSA, acute CPAP therapy may improve outcomes [48]. However, adaptive-servo ventilation (ASV) should be avoided in patients with LV ejection fraction (LVEF) less than 45% and central sleep apnea with concomitant OSA as it increases mortality [49].

## 5. Acute Coronary Syndrome

OSA has been identified as an important risk factor for coronary artery disease (CAD). There is increasing evidence that severe OSA and, less likely, mild-moderate OSA are important contributing factors to cardiovascular complications [50,51,52,53,54,55]. However, due to the lack of randomized controlled studies, it is hard to infer causality from this observed association. In this section, we explore the relationship of OSA with acute coronary syndrome (ACS).

OSA has been correlated with an increased incidence of myocardial infarction, the need for coronary artery interventions, and cardiovascular deaths [56]. Furthermore, the diagnosis of OSA has been associated with poor recovery of LV function in post-MI patients [57]. Multiple observational studies have demonstrated poor outcomes in patients with ACS and OSA. Studies that involved post-percutaneous coronary intervention (PCI) patients have shown that OSA is an important predictor of major adverse cardiovascular and cerebrovascular events (MACCE) [58,59]. In an observational study of 105 patients who presented with ST elevation myocardial infarction and diagnosed with OSA, subsequent follow up at 1.5 years revealed that patients with severe OSA compared to non-severe OSA had increased incidence of cardiovascular adverse events and decreased survival [60].

The mechanisms behind how OSA may lead to atherosclerosis are believed to be multifactorial, and consist of both direct and indirect effects. OSA can increase sympathetic activation, vasoconstriction, oxidative stress, and trigger inflammation that leads to endothelial dysfunction and premature atherosclerosis [61]. In contrast, there is evidence to support the protective effects of OSA. Studies have reported an association of OSA with less severe cardiac injury. A possible explanation of this phenomenon is that OSA is associated with ischemic preconditioning and greater development of coronary artery collateral circulation [61].

However, many recent studies have shown mixed results. In patients with non-sleepy moderate to severe OSA, CPAP treatment when compared to no treatment in a randomized control trial (RCT) did not reduce the incidence of hypertension or new cardiovascular events when followed for a median of four years [62]. Similar to the lack of supportive data for concrete cardiovascular end points in primary prevention trials, the use of PAP therapy has not been shown to change important cardiovascular outcomes for secondary prevention as well. A randomized multicenter controlled trial, the “Effect of obstructive sleep apnea and its treatment with continuous positive airway pressure on the prevalence of cardiovascular events in patients with acute coronary syndrome (ISAACC study)”, investigated the effects of OSA and treatment with CPAP on patients admitted to hospital for ACS [62]. This trial did not show an increased incidence of cardiovascular events in patients with underlying OSA, nor did it show any beneficial effects of utilizing CPAP therapy on MACCE compared to a controlled group, either in the short term or on a median follow-up of 3.35 years, albeit with an improvement of clinical symptoms, such as sleepiness and blood pressure control [62]. This study had multiple limitations and excluded severe OSA with daytime sleepiness and patients with severe ACS with poor prognostic factors, such as shock due to cardiac etiology [62].

In contrast to the ISAACC study, which looked at patients in the acute hospitalized phase, the multicentric RCT, Sleep Apnea Cardiovascular Endpoints study (SAVE), included more than 2700 patients with moderate to severe OSA and established ischemic heart disease and ischemic CVA, had a mean CPAP adherence of 3.3 h every night and an average follow up of 3.7 years. Interestingly, it showed a similar neutral result with primary cardiovascular endpoints (MACCE), despite improvement in OSA symptoms and improvement in quality-of-life measures [63]. In patients who presented with ACS and moderate to severe OSA, post-revascularization, good adherence to CPAP showed improved MACCE outcomes in a single center subgroup of a randomized control study, “Randomized Intervention with Continuous Positive Airway Pressure in CAD and OSA” (RICCADSA) trial, but the results could not be replicated in a subgroup of the larger multicenter randomized control trial ISAACC study [62,64]. Caution needs to be exercised when interpreting these results, as the numbers were small and not directed toward answering the question.

Nevertheless, diagnosing and treating OSA leads to improved clinical symptoms, better BP control, and might have long-term beneficial effects yet to be discovered. Data also suggest that there may be certain phenotypes of OSA that may have a greater benefit of CPAP therapy than others [65]. Observational data from the sleep heart health study showed a more than 40% reduction in mortality in patients who were prescribed PAP therapy and had severe OSA [66]. Randomized controlled data over a longer time duration and involving severe OSA patients alone may help clarify this beneficial signal noted in observational studies [16]. Table 1 lists the limitations of some of these randomized control studies which have studied the effect of positive airway pressure therapy on cardiovascular disease in obstructive sleep apnea.

Further research is needed to answer questions such as the most appropriate timing to evaluate and start treating for OSA in the hospital and differences in outcomes of various OSA phenotypes. In the critical care setting, it is important to recognize OSA as a potential risk factor for adverse outcomes. Some novel risk factors have been described linking OSA and Cardiovascular disease. These are briefly summarized in Figure 1.

## 6. Stroke

Stroke and OSA commonly co-exist. In a meta-analysis involving more than seven thousand patients post-stroke, prevalence of OSA was found to be more than 70% with prevalence of severe OSA being 30% [75]. OSA was persistent through acute phase (less than one month) to chronic phase (more than 3 months from stroke onset) [75]. OSA is associated with new onset and recurrent stroke [76,77]. The proposed mechanisms explaining the increase in this risk possibly include hemodynamic changes during episodic apnea, reduced cerebral blood flow, hypercoagulability, and cerebral ischemia from hypoxia, in addition to other risk factors prevalent in patients with OSA that include arrhythmias such as atrial fibrillation and hypertension [78,79]. CPAP adherence post-stroke is challenging, but once accepted it can help with neurological improvement [80]. A large multi-center randomized controlled study, Sleep for Stroke Management and Recovery Trial (Sleep SMART), may help answer cardiovascular outcomes and neurological function recovery post-stroke and PAP intervention for OSA [76].

## 7. Pulmonary Hypertension

Pulmonary hypertension (PH) is present in about 20% of patients with OSA [81,82]. In some studies, where PH was confirmed by pulmonary artery catheterization, the prevalence of sleep apnea has been found to be as high as 89% [83]. Co-morbid conditions associated with PH and OSA include hypoxia (baseline and nocturnal), hypercapnia, comorbid lung conditions (obstructive more than restrictive lung disease), and obesity-hypoventilation syndrome [81,84]. As the severity of OSA increases, PH also increases [81]. The proposed mechanisms by which OSA can cause PH include hypoxic pulmonary arteriolar vasoconstriction [85] and activation of inflammatory pathways due to chronic sustained hypoxia leading to vascular remodeling and ultimate irreversible increase in pulmonary vascular resistance and right ventricular (RV) dysfunction. Furthermore, intermittent surges in nocturnal blood pressure can also lead to left ventricular hypertrophy and left heart dysfunction, in turn contributing to pulmonary hypertension [85]. PH associated with OSA is mild in the absence of other cardiovascular diseases; however, PH due to a different primary cause can be exacerbated by the presence of OSA [16].

The presence of PH can complicate the care of patients in critical care units. In the setting of PH, various insults can trigger RV failure causing worsening RV function and resulting in poor outcomes and thus, close monitoring is needed to prevent shock and death. Such patients are managed by identifying and correcting triggers, optimizing RV preload while avoiding volume depletion, utilizing systemic vasopressors, and inotropes when appropriate. Some of these patients may require extra corporeal membrane oxygenation (ECMO) in severe cases [85].

Endotracheal intubation and mechanical ventilation should be avoided when feasible, given how small shifts in intrathoracic pressure can dramatically affect RV function and can often lead to poor outcomes for patients with PH and RV dysfunction. Furthermore, induction agents that are often used for intubation can potentially lead to cardiac arrest due to a sudden decrease in systemic vascular resistance and arterial blood pressure [86]. In patients with OSA, the presence of underlying PH should be suspected, and patients should be closely monitored for RV failure.

Isolated OSA causes mild PH, but co-morbid conditions can worsen the degree and outcomes related to PH [16]. When PH and OSA co-exist, this can lead to worse nocturnal oxygen desaturation, decreased exercise capacity, decreased quality of life, and decreased survival at 1, 4, and 8 years [87,88,89]. Treatment with PAP therapy and surgical weight loss can have a modest effect on pulmonary hemodynamics, with evidence on survival outcomes lacking [90,91].

## 8. Thrombo-Embolic Disease

OSA is associated with endothelial dysfunction, platelet dysfunction, and increased coagulability. In a systematic review of fifteen studies, OSA was found to be a risk factor for both pulmonary embolism and deep vein thrombosis to a magnitude of 2–3-fold [92].

## 9. Screening and Treating OSA

Patients with cardiovascular disease should be carefully considered for screening and treating OSA. These include patients with difficult-to-treat hypertension, recurrent or persistent atrial fibrillation, non-AF arrhythmia, pulmonary hypertension, congestive heart failure, and stroke [16]. Treatment recommendations include weight loss, lifestyle changes, PAP therapy in moderate to severe OSA or symptomatic OSA, mandibular advancement therapies in those intolerants of PAP therapy, and neural stimulation techniques [10,16,93]. Other therapies are not well established [16].

## 10. Conclusions

As noted, cardiovascular diseases are frequently associated with OSA. Some of these, such as hypertension, are associated in a negative feedback loop with OSA, in which OSA may worsen hypertension and in turn hypertension may worsen OSA. Treatment of OSA may improve symptoms, quality of life, and help control pulmonary pressures and hypertension modestly. However, good evidence supporting primary and secondary prevention of cardiovascular diseases has not been established. Ongoing clinical trials over a longer time period and involving high risk patients may answer these questions. Nevertheless, it is currently recommended to screen for and treat OSA when cardiovascular diseases are identified.

## Figures and Tables

**Figure 1 medicina-58-01390-f001:**
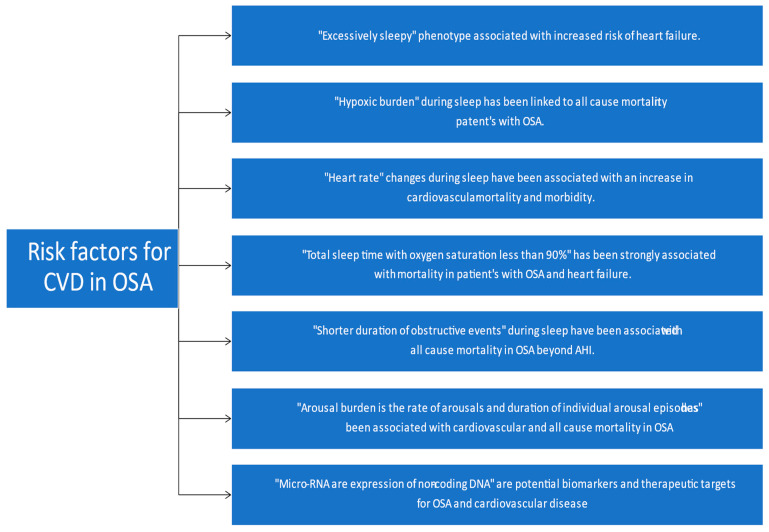
Novel risk factors linking OSA and cardiovascular disease [41,68,69,70,71,72,73,74]. (Modified and adapted from Ref. [68]) OSA: obstructive sleep apnea; AHI: apnea hypopnea index; CVD: Cardiovascular disease.

**Table 1 medicina-58-01390-t001:** Enumerates the limitations of randomized control studies which have investigated the impact of PAP therapy on cardiovascular diseases in OSA [67].

Selection bias	Samples selected were not reflective of the population. Trials recruited patients with established cardiovascular disease and hence were aimed at secondary prevention. Excluded patients with greatest risk of cardiovascular disease, such as patients with “symptomatic sleepiness” due to ethical concerns, hence likely impacting the outcome of the study.
Adherence to PAP therapy was sub optimal.	Patients recruited had established cardiovascular disease and did not present with OSA initially. Hence, adherence to PAP therapy was sub-optimal and invariably less than 4 h/night and hence may have impacted the results of the clinical trials.
Lack of adequate numbers and hence void of statistical power to show difference in outcome	The clinical trials were small in number and lacked statistical power. All the studies had this limitation.
“Composite end point” as outcome was a limitation	The studies chose “Composite end point” as the outcome which consisted of various competing events such as stroke, myocardial infarction, heart failure, angina, etc., which were not equal in terms of weightage. Small sample size, inadequate power and smaller number of individual events altogether impacted the final outcome.

OSA: obstructive sleep apnea; PAP: positive airway pressure.

## Data Availability

Not applicable.

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
