# Peer review of "Cardiovascular Complications of Obstructive Sleep Apnea in the Intensive Care Unit and Beyond"

_medicina, 2022, doi:10.3390/medicina58101390_

Round 1

Reviewer 1 Report

Dear Author,

thank your for interesting paper.

Please express in the title the idea of a narrative review.

Line 51 about craniofacial morphology you could cite this very recent paper

Bertuzzi, F.; Santagostini, A.; Pollis, M.; Meola, F.; Segù, M. The Interaction of Craniofacial Morphology and Body Mass Index in Obstructive Sleep Apnea. Dent. J. 2022, 10, 136. https://doi.org/10.3390/dj10070136

line 252 about mandibular advancement device

Segù M, Campagnoli G, Di Blasio M, Santagostini A, Pollis M, Levrini L. Pilot Study of a New Mandibular Advancement Device. Dent J (Basel). 2022 Jun 6;10(6):99. doi: 10.3390/dj10060099. PMID: 35735642; PMCID: PMC9222002.

Ciavarella D, Campobasso A, Suriano C, Lo Muzio E, Guida L, Salcuni F, Laurenziello M, Illuzzi G, Tepedino M. A new design of mandibular advancement device (IMYS) in the treatment of obstructive sleep apnea. Cranio. 2022 Feb 16:1-8. doi: 10.1080/08869634.2022.2041271. Epub ahead of print. PMID: 35171757.

In chapter 9 it could be very interesting analyze the efficacy of the different treatment modalities in preventing the cardiovascular diseases in OSAS patients

Author Response

Reviewer 1 Comments:

Dear Author,

thank your for interesting paper.

Please express in the title the idea of a narrative review.-we have added the word narrative review in the title as per your comment

Line 51 about craniofacial morphology you could cite this very recent paper

Bertuzzi, F.; Santagostini, A.; Pollis, M.; Meola, F.; Segù, M. The Interaction of Craniofacial Morphology and Body Mass Index in Obstructive Sleep Apnea. Dent. J. 2022, 10, 136. https://doi.org/10.3390/dj10070136

line 252 about mandibular advancement device

Segù M, Campagnoli G, Di Blasio M, Santagostini A, Pollis M, Levrini L. Pilot Study of a New Mandibular Advancement Device. Dent J (Basel). 2022 Jun 6;10(6):99. doi: 10.3390/dj10060099. PMID: 35735642; PMCID: PMC9222002.

Ciavarella D, Campobasso A, Suriano C, Lo Muzio E, Guida L, Salcuni F, Laurenziello M, Illuzzi G, Tepedino M. A new design of mandibular advancement device (IMYS) in the treatment of obstructive sleep apnea. Cranio. 2022 Feb 16:1-8. doi: 10.1080/08869634.2022.2041271. Epub ahead of print. PMID: 35171757.

The above references as suggested by you have been incorporated. Thank you for suggesting.

In chapter 9 it could be very interesting analyze the efficacy of the different treatment modalities in preventing the cardiovascular diseases in OSAS patients- we appreciate the suggestion. We wanted to do this ourselves but did not find enough literature on this.

Appreciate your thoughtfulness and kind consideration.

Reviewer 2 Report

This is an excellent review of the importance of sleep-related breathing disorders associated with other chronic diseases, reviewing the importance of implementing PAP therapy in acute and chronic situations and recognizing the shortcomings of PAP therapy. More studies are needed to understand the impact of PAP therapy on morbidity and mortality.

FYI:  line 69 (atrial, need correction, not a trial )

Line 96  did you mean PAF or AF? 

Good luck!

Author Response

2.Reviewer 2 suggestions:

This is an excellent review of the importance of sleep-related breathing disorders associated with other chronic diseases, reviewing the importance of implementing PAP therapy in acute and chronic situations and recognizing the shortcomings of PAP therapy. More studies are needed to understand the impact of PAP therapy on morbidity and mortality.

FYI:  line 69 (atrial, need correction, not a trial )-Corrected

Line 96  did you mean PAF or AF? =PAF is correct. Study reported on PAF.

Good luck!-Thank you for your kind words. Much appreciated!

Reviewer 3 Report

Dear Authors,

This is an interesting and very well-organized review about OSA and cardiovascular complications, aiming to describe epidemiological and common mechanisms of pathogenesis. However, the title of the manuscript “Cardiovascular Complications of Obstructive Sleep Apnea in the Intensive Care Unit and Beyond” is far from the total body of this review, as referrals about ICU patients with these comorbidities were shown only in the subsection for atrial fibrillation (Lines 77-79). I suggest that the authors reconsider an appropriate title. 

Minor Comment

1.       Lines 157-159. This sentence and the relative reference referred to Central Sleep Apnea and Heart Failure (HF) and not to the coexistence of OSA and HF. The authors have to eliminate this sentence as it may cause confusion to readers. 

Author Response

  1. Reviewer 3
    Dear Authors,

This is an interesting and very well-organized review about OSA and cardiovascular complications, aiming to describe epidemiological and common mechanisms of pathogenesis. However, the title of the manuscript “Cardiovascular Complications of Obstructive Sleep Apnea in the Intensive Care Unit and Beyond” is far from the total body of this review, as referrals about ICU patients with these comorbidities were shown only in the subsection for atrial fibrillation (Lines 77-79). I suggest that the authors reconsider an appropriate title. 

Thank you for your suggestion. We wanted to cover as much as possible for patients with OSA in ICU. However, literature is sparse. Therefore, we extended our review to non-ICU patients involving OSA and cardiovascular complications, with clinical relevance. Recognizing this we have added the word “beyond” in the title. Hope this satisfies. Appreciate your input.

Minor Comment

  1. Lines 157-159. This sentence and the relative reference referred to Central Sleep Apnea and Heart Failure (HF) and not to the coexistence of OSA and HF. The authors have to eliminate this sentence as it may cause confusion to readers. – Thank you for your suggestion. The reason this line has been written is that this was a sentinel study and involved patients with both combined OSA and CSA. Therefore, we believe that this is relevant. ASV mode of ventilation is contra indicated due to harm in patients with CHF and OSA+CSA.
